# Performance of Retro-Reflective Building Envelope Materials with Fixed Glass Beads

**Jihui Yuan** [1,]*[⬤], **Craig Farnham** [2] **and Kazuo Emura** [2]

1   Dept. of Architectural Engineering, Graduate School of Engineering, Osaka University, Suita 565-0871, Japan
2   Dept. of Housing and Environmental Design, Graduate School of Human Life Science, Osaka City
    University, Osaka 558-8585, Japan; farnham@life.osaka-cu.ac.jp (C.F.); emura@life.osaka-cu.ac.jp (K.E.)
*   Correspondence: en@arch.eng.osaka-u.ac.jp; Tel.: +81-6-6879-7645

**Abstract:** Urban heat islands (UHI) are growing in size and intensity, which is partly attributable to the large amount of anthropogenic waste heat. Moreover, heat emitted from building exterior walls accounts for a large portion of the total anthropogenic waste heat. Thus, strategies and technologies for preventing the emission of heat from building exterior walls are being pursued by researchers worldwide. Amongst these technologies, the potential of use of retro-reflective (RR) materials instead of diffuse highly reflective (HR) materials applied to building envelopes for UHI mitigation is being studied widely. However, RR materials haven't been applied to building envelopes in practice due to their unproven weather resistance. In order to develop RR materials with high weather resistance for application to building envelopes, two types of micro glass beads with different refractive indices (1.5, 1.9) and five different colors of base layers were evaluated in this study. Their RR performance was measured by optical experiment and compared to two types of RR sheets commercially available in Japan. The results showed that the glass bead RR samples with a refractive index of 1.9 had much higher retro-reflectivity (better RR capacity) compared to those with a refractive index of 1.5.

**Keywords:** urban heat island; retro-reflective materials; glass beads; refractive index; optical analysis; retro-reflectivity

## 1. Introduction

The urban heat island (UHI) phenomenon is a well-documented climatic change phenomenon in large cities worldwide [1,2]. The temperature differences between urban regions and surrounding rural regions are continuing to rise due to the UHI phenomenon and climate change [3], especially in the summer period. This might lead to an increase in building energy consumption during the summer period of cooling demand and affect the quality of human life [4,5]. Thus, many strategies of mitigating UHI are being carried out globally.

Among these strategies for UHI mitigation, highly reflective (HR) building envelopes are considered as one solution for UHI mitigation and building energy savings [6]. Research has indicated that HR materials could reduce the surface temperature of buildings, with peak drop up to 11.5 °C [7]. "Cool roof" HR membranes applied to building rooftops have large potential in reducing surface temperature and energy savings [8]. Several inorganic materials applied to building exterior walls have been developed for improving the built environment. X-ray diffraction and differential thermal analysis were applied to verify the composition of these developed materials. The optical performance was evaluated through surface temperature measurement in the outdoor environment [9].

Most of these HR coating materials noted above are applied to building rooftops. Some researchers have suggested that it is possible to replace the HR coating materials with retro-reflective (RR) materials as building coating materials to mitigate the UHI effect, since the RR materials can reflect the incident

sunlight back towards the sky and will give less of an effect to the surrounding buildings and roads [10–12]. Currently, however, these RR materials are usually only employed for various safety and decorative purposes, e.g., traffic signs. They have not been applied to building exterior wall surfaces due to unknown durability and high cost of these RR materials. Therefore, there is interest in research on making the application of RR materials to building facades more practical and cost-effective.

Furthermore, the retro-reflectivity typically varies depending on the incident angle of light, from perpendicular to the material (0°) to just before becoming parallel (at 90°). RR materials intended for traffic signs and safety clothing are desired to have the best retro-reflectance at low angles. Traffic signs must reflect strongly at oncoming vehicles. Strong retro-reflectivity at high angles is not a concern. However, for application to building surfaces to reflect sunlight, good reflectivity at high incidence angles is important. Midday sunlight on summer days can have incidence angles relative to vertical walls up to 90°, depending on building latitude and solar position. Early morning and late afternoon sun may also have such high incidence angles reflecting light east-west in summer. Conversely, a sunlight incidence angle near zero is quite rare for vertical walls, but common at midday for roofs. For RR materials to be effective throughout the day as a building-surface UHI countermeasure, they should be effective over a wide range of incidence angles, rather than the narrower range needed for traffic signs and the like.

In the current material market, RR materials are commonly made with one of two reflective components, glass beads or prisms. A simulation analysis of glass beads with different refractive indices of 1.5, 1.9, and 2.2 was carried out in our previous research [13]. It was shown that glass beads with refractive index of 1.9 were the most effective to mitigate UHI effect as RR envelope materials, compared to those with refractive indices of 1.5, 1.9, and 2.2. The RR performance of several glass bead-type RR materials has been investigated using an optical apparatus [14,15]. It was shown that the RR capacity of these RR materials becomes weaker as the incidence angle of light is increased from low to high values, and the specular (mirror) reflection becomes stronger. It was concluded that the glass bead-type RR materials are more effective when the incidence angle of sunlight is low. Several RR materials using glass beads with refractive indices of 1.5 and 1.9 have been developed by a research [16]. The reflective directional characteristics of these developed glass bead RR materials were investigated using optical apparatus in the laboratory. It showed that the RR material with a refractive index of 1.9 has the best RR capacity in terms of mitigating UHI effect and saving building energy. In addition, the difference between building envelope samples using glass microspheres and the same sample without the glass microspheres was compared in terms of retro-reflectivity at different incidence of light. It showed that the envelope samples without glass microspheres have almost the same overall solar reflectivity, however, the retro-reflectivity at different incidence is very small compared to the envelope samples with glass microspheres. Thus, it was concluded that the glass microspheres have a positive effect in the development of RR materials. New ceramic colored tiles using RR microspheres have been developed, and the continuous industrial process of producing the new RR tiles in an economic and sustainable way was introduced in a research [17]. The optical performance of these RR ceramic colored tiles was evaluated by spectrophotometric, angular reflectance and colorimetric analysis. Research investigated and compared the performance of three materials used for external wall surfaces, including a common ceramic commercial tile, a glass tile obtained by covering the ceramic tile with a transparent paint on which glass spheres are spread, and a barium tile obtained by spreading treated barium microspheres on the wet transparent paint on the ceramic tile [18]. The optic analysis by spectrophotometer indicated that the barium tile has the highest overall solar reflectivity (39%), compared to the common ceramic tile (30%) and the glass tile (32%). The angular reflectance analysis showed that both the glass tile and the barium tile have stronger RR behavior for most of the incident directions, compared to the common ceramic tile. The analysis on a potential in UHI reduction showed that both the glass tile and the barium tile are more effective to mitigate the UHI effect, compared to the common ceramic tile.

The RR performance of three-mirror and four-mirror types of corner RR samples that are another-type RR materials (different from glass microsphere-type) had been analyzed using geometrical optical principles in the research [10]. A type of prism RR material coated with a type of glass sheet was developed [11]. The durability of the developed prism RR materials was investigated by thermal measurement for a long-term exposure (about 1.5 years) in the outdoor environment. It showed that both the solar reflectivity and retro-reflectivity of the prism RR material showed no significant decrease during the exposure period of about 1.5 years. Additionally, a simulation analysis of different reflective characteristics on the urban albedo was implemented. It showed that the prism RR envelope applied to simulated buildings in an urban canyon has the largest potential in terms of mitigating the UHI effect, compared to diffuse reflective (DR) and mirror reflective (MR) envelopes.

Based on the above studies of RR materials for possible application to building facades to fight the UHI phenomenon, this study aims to use the glass beads to develop some glass bead-type RR samples and evaluate their retro-reflectivity for different incident angles.

## 2. Experimental Materials

### 2.1. Glass Bead RR Samples

As detailed in Table 1, a total of 10 glass bead RR samples (100 mm square galvanized steel plates) with a refractive index of 1.5 or 1.9 and a reflective layer of white, silver, yellow, gray, or transparent acrylic paint were made in this research. In order to compare the RR capacity of these developed glass bead RR samples with that of the RR material commercially available in the material market, two types of RR sheets (capsule and prism sheets) were chosen in this study. The surface appearance of these RR samples is shown in Figure 1, with the two commercial RR materials at bottom-left. The structure of developed glass bead RR samples is shown in Figure 2.

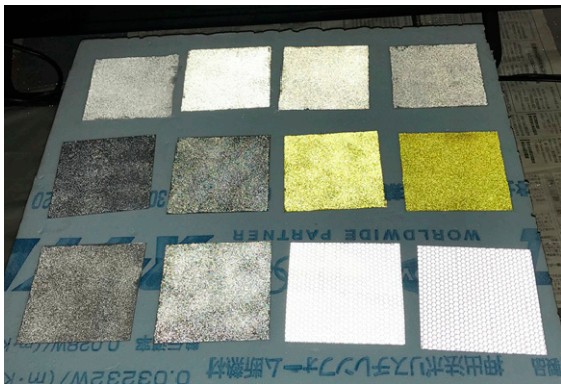

**Figure 1.** Appearance of 10 developed glass bead retro-reflective (RR) samples and two types of RR sheets.

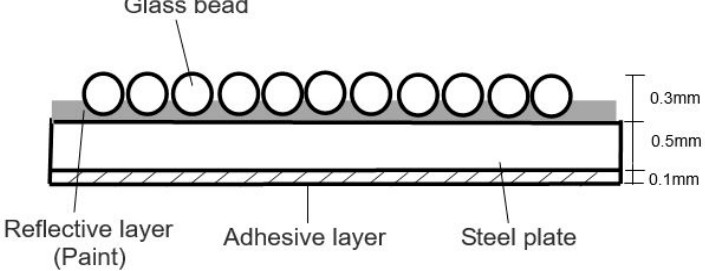

**Figure 2.** Structure of developed glass bead RR samples.

**Table 1.** Characteristics of 10 developed glass bead RR samples and two types of RR sheet.

| Samples | Reflective Layer | Refractive Index of Glass Bead [-] | Diameter of Glass Beads (μm) | Density of Glass Beads (kg/m$^2$) |
|---|---|---|---|---|
| Sample 1 | white | 1.5 | 106–850 | 0.30 |
| Sample 2 | white | 1.9 | 106–850 | 0.30 |
| Sample 3 | yellow | 1.5 | 106–850 | 0.30 |
| Sample 4 | yellow | 1.9 | 106–850 | 0.30 |
| Sample 5 | gray | 1.5 | 106–850 | 0.30 |
| Sample 6 | gray | 1.9 | 106–850 | 0.30 |
| Sample 7 | silver | 1.5 | 106–850 | 0.30 |
| Sample 8 | silver | 1.9 | 106–850 | 0.30 |
| Sample 9 | transparent | 1.5 | 106–850 | 0.30 |
| Sample 10 | transparent | 1.9 | 106–850 | 0.30 |
| Capsule | | - | | |
| Prism | | - | | |

## 2.2. Optical Apparatus in the Laboratory

The optical apparatus, which is also called the "emitting-receiving optical fiber system", was assembled in the laboratory and is shown in Figure 3. It comprises spectrophotometers, including a visible (VIS) spectrophotometer and a near-infrared (NIR) spectrophotometer; a halogen lamp light source; glass fiber with an optical fiber probe; sample stage; angle adjustment; and a computer for processing data. The distance from the optical fiber probe to the RR sample surface is set to 30 mm because the incident light can be well focused on the surface of samples in this study.

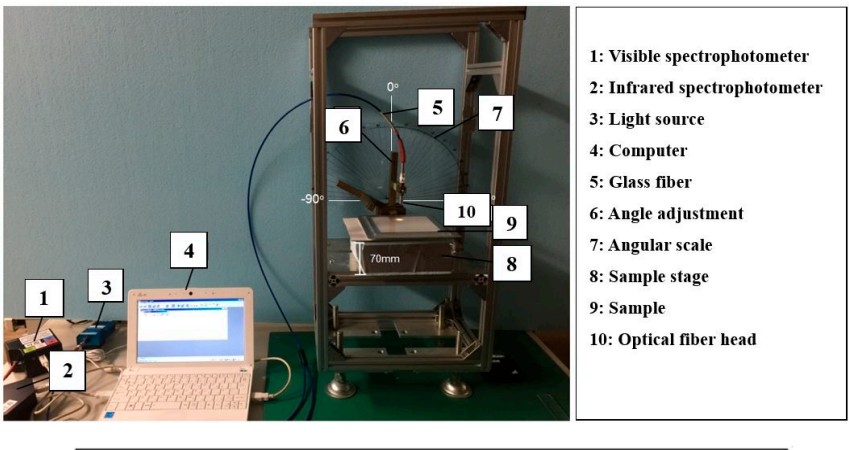

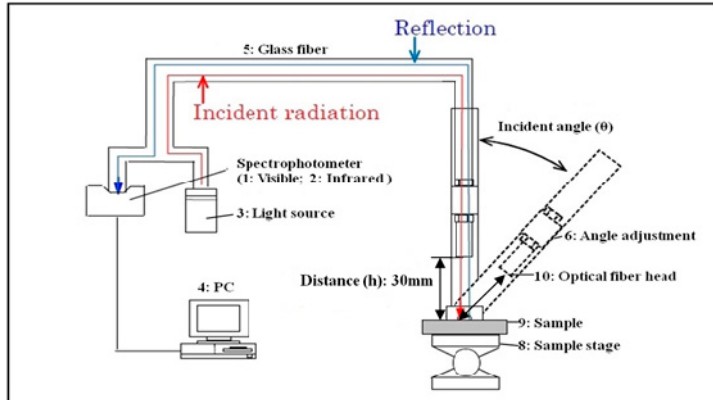

**Figure 3.** Emitting-receiving optical fiber system.

## 3. Methodology for Evaluating Angular Retro-Reflectivity

In order to evaluate the angular retro-reflectivity of these developed RR samples, the method that was proposed in our previous study [12] was adopted in this paper. Note that the apparatus used in the method used here creates angular retro-reflectivity profiles relative to a chosen RR sample, rather than directly measuring absolute retro-reflectivity coefficients.

The function for determining the angular retro-reflectivity of RR samples is shown in the following Equation (1),

$$\rho_{\text{ret-ang}} = \rho_{\text{ret-ref}} \times \int S(\lambda) \times E(\lambda)d\lambda / \int S_{\text{ref}}(\lambda) \times E(\lambda)d\lambda \tag{1}$$

where "$\rho_{\text{ret-ang}}$" is the angular retro-reflectivity of RR samples, "$\rho_{\text{ret-ref}}$" is the retro-reflectivity with incident angle of 7° of a commercially-available prism RR sheet as reference (its absolute retro-reflectivity is approximately 0.41 according to the previous study [19]), "$S(\lambda)$" is the reflection intensity of these samples at the chosen incident angle, "$S_{\text{ref}}(\lambda)$" is the reflection intensity of the prism RR sample at an incident angle of 7°, and "$E(\lambda)$" is the spectral distribution of hemispherical solar irradiance specified in ISO 9845-1 of the International Organization for Standardization [20].

## 4. Results and Discussion

### 4.1. Angular Retro-Reflectivity of RR Samples

The angular retro-reflectivity of 10 developed RR samples and two types of RR materials (capsule sheet and prism sheet) commercially available in the market was evaluated and is detailed in Figure 4. It was found that the prism sheet has the largest retro-reflectivity of approximately 0.41 at the incident angle of 0°, and its retro-reflectivity decreased as the incident angle was increased from 0 to 85° (the retro-reflectivity decreased sharply from the incident angle of about 75°). The capsule sheet had a similar change in retro-reflectivity to the prism sheet. Its retro-reflectivity decreased sharply from the incident angle of 70°. Compared to the prism and capsule sheets, samples 2, 4, and 6 (white, yellow and gray) have a relatively smaller angular retro-reflectivity of approximately 0.27 at the incident angle of 0° and have a relatively stable change in angular retro-reflectivity until the incident angle is increased to 80°. Sample 10 (transparent paint) has a relatively larger retro-reflectivity of about 0.28 at the incident angle of 0°, however, its retro-reflectivity decreased sharply from the incident angle of 30°.

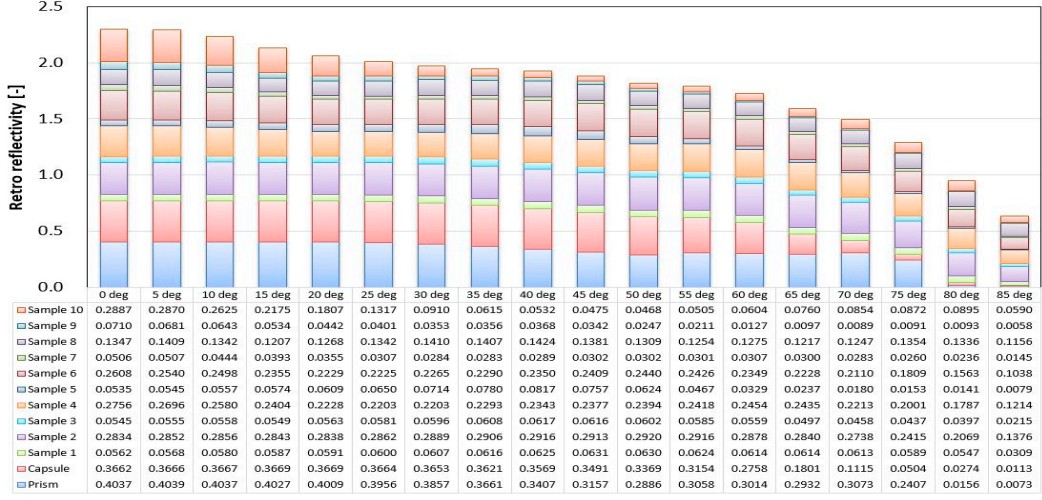

| | 0 deg | 5 deg | 10 deg | 15 deg | 20 deg | 25 deg | 30 deg | 35 deg | 40 deg | 45 deg | 50 deg | 55 deg | 60 deg | 65 deg | 70 deg | 75 deg | 80 deg | 85 deg |
|---|---|---|---|---|---|---|---|---|---|---|---|---|---|---|---|---|---|---|
| Sample 10 | 0.2887 | 0.2870 | 0.2625 | 0.2175 | 0.1807 | 0.1317 | 0.0910 | 0.0615 | 0.0532 | 0.0475 | 0.0468 | 0.0505 | 0.0604 | 0.0760 | 0.0854 | 0.0872 | 0.0895 | 0.0590 |
| Sample 9 | 0.0710 | 0.0681 | 0.0643 | 0.0534 | 0.0442 | 0.0401 | 0.0353 | 0.0356 | 0.0368 | 0.0342 | 0.0247 | 0.0211 | 0.0127 | 0.0097 | 0.0089 | 0.0091 | 0.0093 | 0.0058 |
| Sample 8 | 0.1347 | 0.1409 | 0.1342 | 0.1207 | 0.1268 | 0.1342 | 0.1410 | 0.1407 | 0.1424 | 0.1381 | 0.1309 | 0.1254 | 0.1275 | 0.1217 | 0.1247 | 0.1354 | 0.1336 | 0.1156 |
| Sample 7 | 0.0506 | 0.0507 | 0.0444 | 0.0393 | 0.0355 | 0.0307 | 0.0284 | 0.0283 | 0.0289 | 0.0302 | 0.0302 | 0.0301 | 0.0307 | 0.0300 | 0.0283 | 0.0260 | 0.0236 | 0.0145 |
| Sample 6 | 0.2608 | 0.2540 | 0.2498 | 0.2355 | 0.2229 | 0.2225 | 0.2265 | 0.2290 | 0.2350 | 0.2409 | 0.2440 | 0.2426 | 0.2349 | 0.2228 | 0.2110 | 0.1809 | 0.1563 | 0.1038 |
| Sample 5 | 0.0535 | 0.0545 | 0.0557 | 0.0574 | 0.0609 | 0.0650 | 0.0714 | 0.0780 | 0.0817 | 0.0757 | 0.0624 | 0.0467 | 0.0329 | 0.0237 | 0.0180 | 0.0153 | 0.0141 | 0.0079 |
| Sample 4 | 0.2756 | 0.2696 | 0.2580 | 0.2404 | 0.2228 | 0.2203 | 0.2203 | 0.2293 | 0.2343 | 0.2377 | 0.2394 | 0.2418 | 0.2454 | 0.2435 | 0.2213 | 0.2001 | 0.1787 | 0.1214 |
| Sample 3 | 0.0545 | 0.0555 | 0.0558 | 0.0549 | 0.0563 | 0.0581 | 0.0596 | 0.0608 | 0.0617 | 0.0616 | 0.0602 | 0.0585 | 0.0559 | 0.0497 | 0.0458 | 0.0437 | 0.0397 | 0.0215 |
| Sample 2 | 0.2834 | 0.2852 | 0.2856 | 0.2843 | 0.2838 | 0.2862 | 0.2889 | 0.2906 | 0.2916 | 0.2913 | 0.2920 | 0.2916 | 0.2878 | 0.2840 | 0.2738 | 0.2415 | 0.2069 | 0.1376 |
| Sample 1 | 0.0562 | 0.0568 | 0.0580 | 0.0587 | 0.0591 | 0.0600 | 0.0607 | 0.0616 | 0.0625 | 0.0631 | 0.0630 | 0.0624 | 0.0614 | 0.0614 | 0.0613 | 0.0589 | 0.0547 | 0.0309 |
| Capsule | 0.3662 | 0.3666 | 0.3667 | 0.3669 | 0.3669 | 0.3664 | 0.3653 | 0.3621 | 0.3569 | 0.3491 | 0.3369 | 0.3154 | 0.2758 | 0.1801 | 0.1115 | 0.0504 | 0.0274 | 0.0113 |
| Prism | 0.4037 | 0.4039 | 0.4037 | 0.4027 | 0.4009 | 0.3956 | 0.3857 | 0.3661 | 0.3407 | 0.3157 | 0.2886 | 0.3058 | 0.3014 | 0.2932 | 0.3073 | 0.2407 | 0.0156 | 0.0073 |

**Figure 4.** Change in angular retro-reflectivity of 10 developed RR samples and two RR materials commercially available in Japan.

It is shown that the samples 1, 3, 5, 7, and 9 (beads with a refractive index of 1.5) have smaller retro-reflectivity compared to the other RR samples.

Moreover, the developed RR samples with the same color layer and different refractive index of 1.5 or 1.9 were compared and shown in Figures 5–10. It was found that the angular retro-reflectivity of samples with a refractive index of 1.9 is larger (often 2-5 times higher or more) than that of samples with refractive index of 1.5 applied to the same color reflective layer.

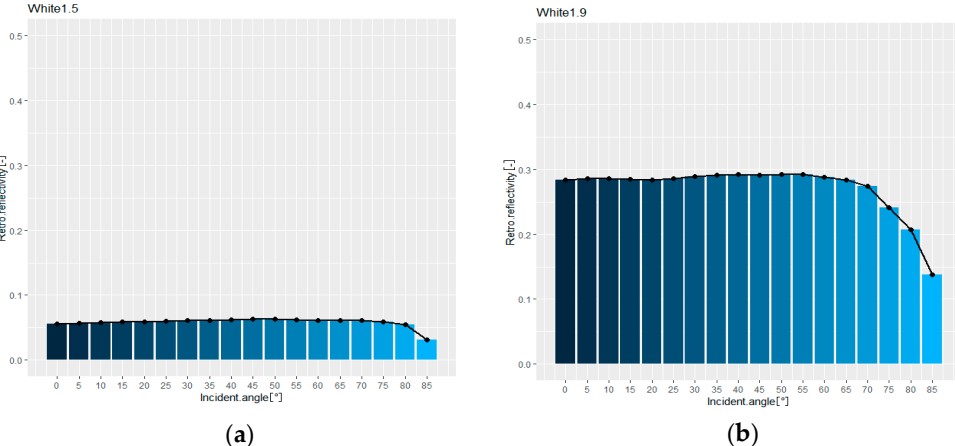

**Figure 5.** Angular retro-reflectivity of RR sample 1, (**a**) white layer and refractive index of 1.5 and RR sample 2 (**b**) white layer and refractive index of 1.9.

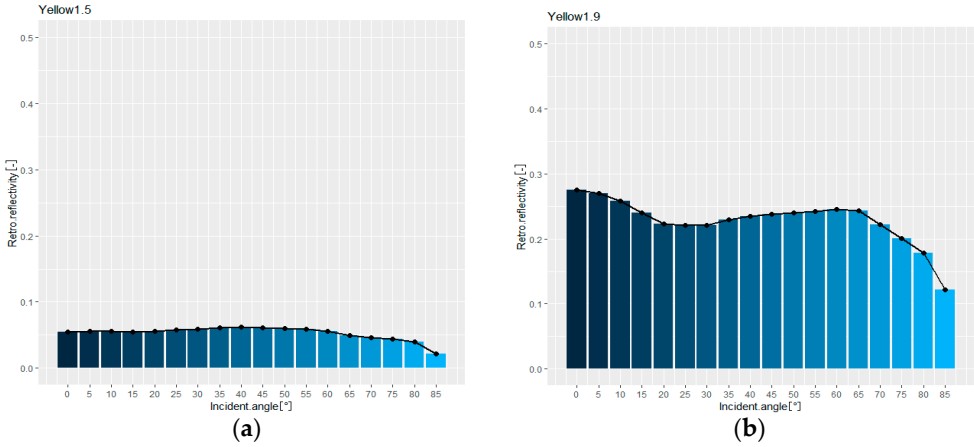

**Figure 6.** Angular retro-reflectivity of RR sample 3, (**a**) yellow layer and refractive index of 1.5 and RR sample 4 (**b**) yellow layer and refractive index of 1.9.

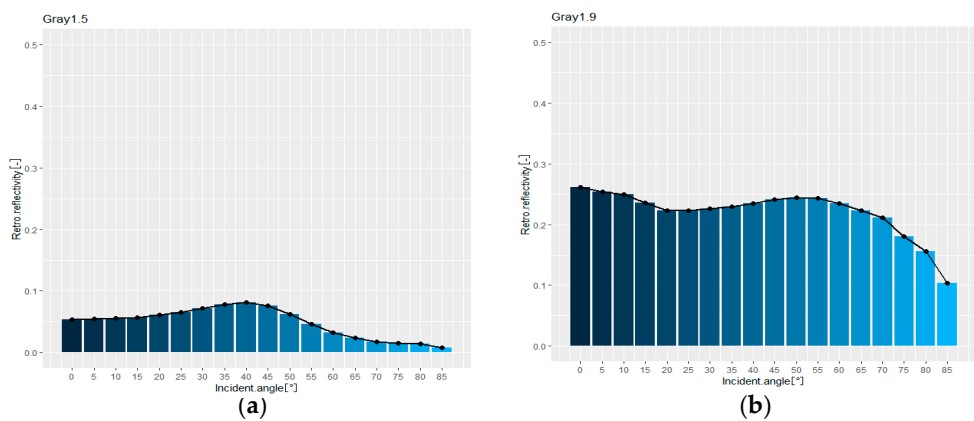

**Figure 7.** Angular retro-reflectivity of RR sample 5 (**a**) gray layer and refractive index of 1.5 and RR sample 6 (**b**) gray layer and refractive index of 1.9.

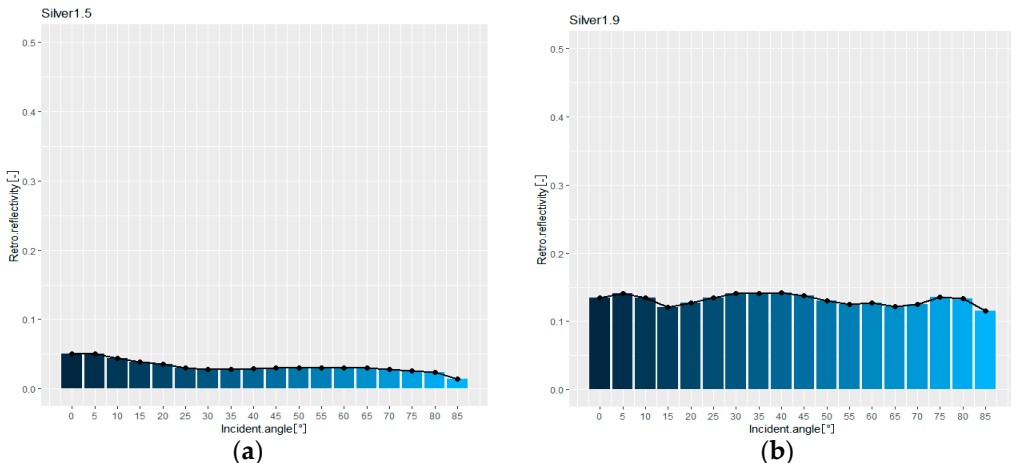

**Figure 8.** Angular retro-reflectivity of RR sample 7 (**a**) silver layer and refractive index of 1.5 and RR sample 8 (**b**) silver layer and refractive index of 1.9.

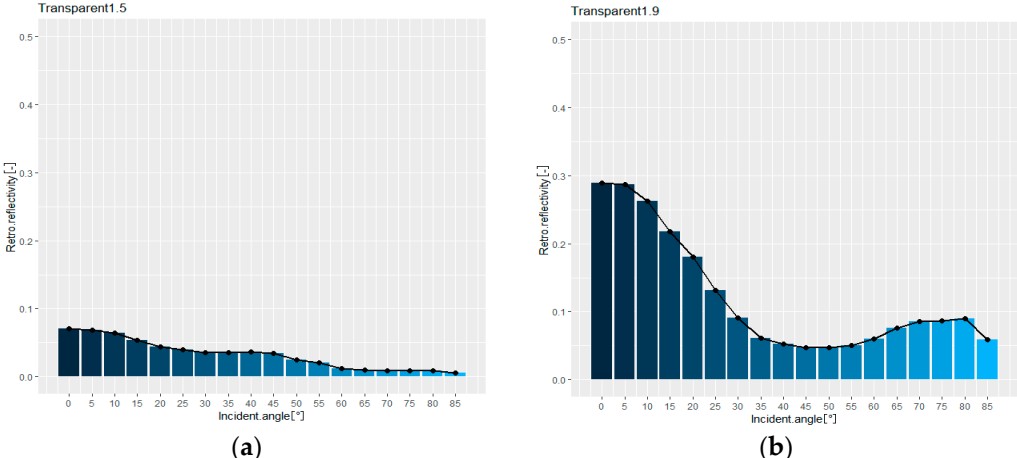

**Figure 9.** Angular retro-reflectivity of RR sample 9 (**a**) transparent layer and refractive index of 1.5 and RR sample 10 (**b**) transparent layer and refractive index of 1.9.

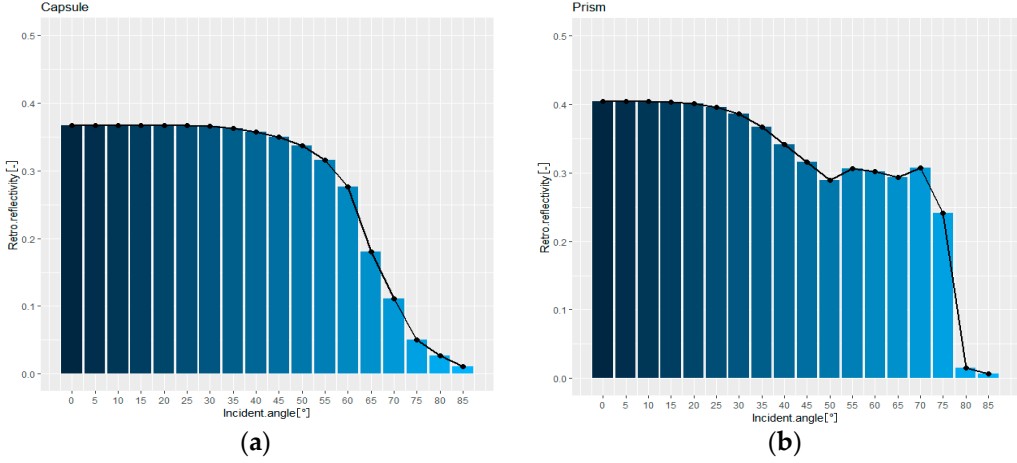

**Figure 10.** Angular retro-reflectivity of RR sample (**a**) capsule sheet and RR sample (**b**) prism sheet.

*4.2. Discussion*

From the above results, we can see that the glass beads with a refractive index of 1.9 are more effective than 1.5 in improving the retro-reflectivity of a material surface. The commercial prism sheet

and capsule sheet have larger retro-reflectivity than the developed RR samples at low incident angles (from 0° to 60°). However, their price is relatively expensive, and the retro-reflectivity is lower than some of the developed samples when the incident angle is above 60° (for the capsule RRM) or 75° (for the prism RRM). The angular retro-reflectivity of sample 2 (white layer and glass bead with a refractive index of 1.9), sample 4 (yellow layer and glass bead with a refractive index of 1.9) and sample 6 (gray layer and glass bead with a refractive index of 1.9) are larger than the other developed glass bead RR samples but still lower than the prism and capsule sheet for the 0° to 60° incidence angles. Moreover, their change in angular retro-reflectivity is relatively constant over the range of angles, and the price of glass beads is relatively cheap compared to the prism and capsule sheet. Further, glass beads can be applied to wet paint without specialist knowledge or equipment. Therefore, it is considered that a white, yellow, or gray layer and glass bead with a refractive index of 1.9 could possibly be used to create the RR materials for building surfaces and could be better than the other color layers and glass beads with a refractive index of 1.5 through this experiment.

These findings are in accordance with our previous research [13]. Thus, it is considered that the glass bead RR material with a refractive index of 1.9 is more effective than that with a refractive index of 1.5 in terms of reflecting incident light back towards the source when applied to building facades as a RR material at lower incident angles. This would make them better at retro-reflecting sunlight throughout the day.

## 5. Conclusions

In this study, a total of 10 glass bead RR samples with different refractive indices (1.5 or 1.9) and different color reflective layers (white, yellow, gray, silver, transparent) were developed. In order to compare these developed glass bead RR samples to RR materials commercially available in the market, two types of RR materials were chosen in this study. The angular retro-reflectivity of these developed glass bead RR samples was evaluated by using an emitting-receiving optical fiber system in the laboratory.

The major points of this research are:

- The prism and capsule sheet have a relatively higher angular retro-reflectivity than the other developed glass bead RR samples, however, their angular retro-reflectivity decreased sharply when the incident angle of light is below about 60° (capsule) or 75° (prism). When the incident angle is increased to 80°, their retro-reflectivity is nearly zero (about 0.02). Thus, it is considered that the prism and capsule sheet commercially available in the market are not effective for retro-reflecting incident sunlight at high incident angles.

- Compared to the prism and capsule sheet, the change in angular retro-reflectivity of the developed glass bead RR samples with white, yellow, gray layers and a refractive index of 1.9 is relatively stable when the incident angle is varied from low to high values. However, the angular retro-reflectivity of these developed glass bead samples is a bit smaller (average value of about 0.13) than that of commercially-available prism and capsule RR sheets. Thus, it is considered that it is possible to use the glass bead with a refractive index of 1.9 and white, yellow, or gray layer to create the RR samples instead of the expensive prism and capsule sheet commercially available in the market.

- Among developed glass bead RR samples with five different colors of reflective layers and two refractive indices, the results showed that RR samples with white, yellow and gray reflective layers and a refractive index of 1.9 have better RR performance and relatively constant change in angular retro-reflectivity over the range of angles than those with silver and transparent reflective layers and a refractive index of 1.5.

- No matter what color reflective layers, the developed glass bead RR samples with a refractive index of 1.5 have smaller angular retro-reflectivity than that with a refractive index of 1.9. Thus, it is considered that the glass bead with a refractive index of 1.9 is more effective in mitigating the UHI effect as a RR material rather than that with a refractive index of 1.5.

For further work, these glass bead RR materials will be considered for industrial production and will be applied to simulated or actual building facades. The effect of the glass bead RR materials on the indoor thermal comfort and outdoor environment will also be evaluated in the future.

**Author Contributions:** J.Y. and C.F. created the materials; J.Y. implemented the optical experiment; J.Y. and K.E. analyzed the data of the experiment; J.Y. wrote the paper; and C.F. checked the English.

**Funding:** This research received no external funding.

**Acknowledgments:** The authors are sincerely grateful to the UNITIKA Corporation of Japan for providing the glass beads used to create the RR samples.

**Conflicts of Interest:** The authors declare no conflict of interest.

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
