# Peer review of "Performance of Retro-Reflective Building Envelope Materials with Fixed Glass Beads"

_applsci, doi:10.3390/app9081714_

Round 1
Reviewer 1 Report
1. I think that this sentence “Especially in the summer period, the urban temperature differences from surrounding rural regions are continuing to rise due to the UHI phenomenon and climate change” (l 30-32) should be rewritten because is not clear.
2. The results of the cited article [5] showed that “total energy consumption is decreased by the effects of the UHI; decreases in space-heating and water-heating energy consumption were larger than the increase in cooling-energy consumption. Therefore, although many proposals for mitigation of UHIs have been offered, these measures might result in an increase in energy consumption”. Therefore, the statement on l 32-34 “This leads to an increase in building energy consumption for cooling demand and affects the quality of human life [4,5]. Thus, many strategies of mitigating UHI are being carried out globally” is not entirely supported by the cited article [5].
3. It appears to me that the cited article [6] does not support the statement made on l 36-37, “Research has indicated that HR materials could reduce the surface temperature of buildings by about 10ᵒC [6]”.
4. I have not found in the cited article [12] reference to the study of the refractive index of 1.7, as it is described.
5. In Figure 3, the numbers in the equipment picture and the caption are too small, they should be increased.
6. The reference [19] “ISO 9050: Glass in building -Determination of light transmittance, solar direct transmittance, total solar energy transmittance, ultraviolet transmittance and related glazing factors (2003)” is not the same as the one cited in l 146 “ISO 9845-1 of International Organization for Standardization [19]”.
7. In the statement made on l 154-156, “Compared to the prism and capsule sheets, Samples 2, 4, 6 (white, yellow and gray) have a relatively smaller angular retro-reflectivity of approximately 0.27”, it seems to me that the 0.27 is referring to the incident angle of 0 degrees so that should be written.
8. In Figure 4 the numbers should be increased in order to make reading easier. Also, the figure could be increased.
9. The paragraph on l 209-211 is redundant in my opinion because it is already written in l 202-205. Also, instead of describing “The major points of this research” from l 208 onwards, it is more important, as we are in the conclusions, to write the main conclusions achieved so it would be ok to take out the statement made in l 209-211.
10. There is nothing mentioned in the conclusions about the comparison of the results between the five different colours of base layers evaluated in this study. Also, the main result written in the abstract of this article in l 22-24 “The results showed that the glass bead RR samples with a refractive index of 1.9 has much higher retro-reflectivity (better RR capacity), compared to those with a refractive index of 1.5” was already obtained in the cited article [15], of 2016.
11. Overall, I think there is value in the research provided in this article.
Author Response
1. I think that this sentence “Especially in the summer period, the urban temperature differences from surrounding rural regions are continuing to rise due to the UHI phenomenon and climate change” (l 30-32) should be rewritten because is not clear.
Response: We have rewritten it accordingly.
2. The results of the cited article [5] showed that “total energy consumption is decreased by the effects of the UHI; decreases in space-heating and water-heating energy consumption were larger than the increase in cooling-energy consumption. Therefore, although many proposals for mitigation of UHIs have been offered, these measures might result in an increase in energy consumption”. Therefore, the statement on l 32-34 “This leads to an increase in building energy consumption for cooling demand and affects the quality of human life [4,5]. Thus, many strategies of mitigating UHI are being carried out globally” is not entirely supported by the cited article [5].
Response: Thank you for your valuable comments. We want to express that although the UHI phenomenon might decrease the total energy consumptions of year in some cities, the UHI phenomenon still leads to increasing the energy consumption of cooling period and largely affect the quality of human life in hot summer. In addition, the energy consumption of cooling period is increased due to UHI effect has also been stated in reference [5].
We have revised this sentence accordingly.
3. It appears to me that the cited article [6] does not support the statement made on l 36-37, “Research has indicated that HR materials could reduce the surface temperature of buildings by about 10ᵒC [6]”.
Response: Thank you for your valuable comment. We have revised this part and cited the other reference of [7] into paper.
4. I have not found in the cited article [12] reference to the study of the refractive index of 1.7, as it is described.
Response: Thank you for your important findings. We wrote the “1.7” by mistake and have deleted the “1.7”.
5. In Figure 3, the numbers in the equipment picture and the caption are too small, they should be increased.
Response: We have improved the Fig.3 accordingly.
6. The reference [19] “ISO 9050: Glass in building -Determination of light transmittance, solar direct transmittance, total solar energy transmittance, ultraviolet transmittance and related glazing factors (2003)” is not the same as the one cited in l 146 “ISO 9845-1 of International Organization for Standardization [19]”.
Response: Thank you for your important findings. We accidentally wrote the wrong document and have corrected the reference [19].
7. In the statement made on l 154-156, “Compared to the prism and capsule sheets, Samples 2, 4, 6 (white, yellow and gray) have a relatively smaller angular retro-reflectivity of approximately 0.27”, it seems to me that the 0.27 is referring to the incident angle of 0 degrees so that should be written.
Response: Thank you for your valuable advice. We have corrected it accordingly.
8. In Figure 4 the numbers should be increased in order to make reading easier. Also, the figure could be increased.
Response: Thank you for your valuable advice. Considering the structure and space of paper, please allow us to change the font size in Fig.4 appropriately to make reading much easier.
9. The paragraph on l 209-211 is redundant in my opinion because it is already written in l 202-205. Also, instead of describing “The major points of this research” from l 208 onwards, it is more important, as we are in the conclusions, to write the main conclusions achieved so it would be ok to take out the statement made in l 209-211.
Response: Thank you for your valuable advice. We have taken out the statement written in line 209-211 accordingly.
10. There is nothing mentioned in the conclusions about the comparison of the results between the five different colours of base layers evaluated in this study. Also, the main result written in the abstract of this article in l 22-24 “The results showed that the glass bead RR samples with a refractive index of 1.9 has much higher retro-reflectivity (better RR capacity), compared to those with a refractive index of 1.5” was already obtained in the cited article [15], of 2016.
Response: Thank you for your valuable comment. We have added the statements on the comparison among five different color reflective layers in the conclusions. The RR materials created in Reference [15] have glass sheet covering, and they are different from the developed RR materials in this study. Thus, it is necessary to state the results “glass bead RR samples with a refractive index of 1.9 also has much higher retro-reflectivity (better RR capacity), compared to those with a refractive index of 1.5” for new materials.
11. Overall, I think there is value in the research provided in this article.
Response: Thank you for your positive comment. We have improved this paper accordingly.

Reviewer 2 Report
The paper needs to undergo some very thorough editing. Presently, there are many language usage errors that are having a major impact on the clarity.
In the abstract, the authors present the objective of the paper as being to develop RR materials with a low cost premium and high weather resistance. However, the paper's content and conclusion do not follow from these objectives. It would seem that the paper primarily sought to investigate the retro-reflectivity of micro glass beads with different refractive indices. The content and conclusion must be seen to follow from the objectives.
Please, be very clear on the objectives
The content is very lacking on the cost aspects of the RR materials. It is important that a thorough review of the cost aspects be undertaken. Further, for comparisons between the proposed materials and the commercially available ones, a clear benchmark needs to be set and explained accordingly.
Line 35,36,41,42 - Please provide sources for the claims made therein.
Line 43, is HR the same thing as DHR? Please be consistent in the use of acronyms.
Lines 65-77 suggest that a study with the exact same objective as this present paper has already been conducted by Reference Sources 12 and 15. Please, be very clear on the contributions that the present paper seeks to make. Otherwise, the paper's novelty becomes very questionable.
Section 2.1, please provide some logistical and/or methodological justification for choosing the colours.
Figure 2 needs to be drawn to scale. To be sure, 3mm can not look exactly as 0.3mm.
Line 130, please provide contextual justification for using the said 30mm distance.
Following the methodology for evaluating retro-reflectivity in section 3, the paper needs to provide a methodology for evaluating the cost performance of the proposed materials relative to the commercially available materials.
Line 164, which ones are the 'developed RR samples'?
Line 181&182 - It would appear that this claim was already made by reference sources 12 and 15. Are you simply replicating their work?
Author Response
Response to Reviewer 2
The paper needs to undergo some very thorough editing. Presently, there are many language usage errors that are having a major impact on the clarity.
In the abstract, the authors present the objective of the paper as being to develop RR materials with a low cost premium and high weather resistance. However, the paper's content and conclusion do not follow from these objectives. It would seem that the paper primarily sought to investigate the retro-reflectivity of micro glass beads with different refractive indices. The content and conclusion must be seen to follow from the objectives.
Please, be very clear on the objectives
Response: Thank you for your valuable comment. We have deleted the contents on the cost of materials and revised the abstract, because the price of glass beads used for developing RR samples in this study hasn’t yet been disclosed by collaborative research company. The RR materials (prism and capsule sheet) commercially available in market are relatively expensive and they are only applied to traffic signs and haven’t been used to building coatings. We will show the results of cost savings between developed RR samples and commercial available ones in market in our future work.
The content is very lacking on the cost aspects of the RR materials. It is important that a thorough review of the cost aspects be undertaken. Further, for comparisons between the proposed materials and the commercially available ones, a clear benchmark needs to be set and explained accordingly.
Response: Thank you for your valuable comment. We will do the cost aspects of RR materials in the future work.
Line 35,36,41,42 - Please provide sources for the claims made therein.
Response: Thank you for your valuable comment. We added the reference for the claim in line 35-36. The claim in line 39-42 “Several inorganic materials applied to building exterior walls have been developed for improving the built environment. X-ray diffraction and differential thermal analysis were applied to verify the composition of these developed materials. The optical performance was evaluated through surface temperature measurement in the outdoor environment” is from reference [9].
. It was concluded that the use of cool non-aged asphalt can reduce the ambient temperature by up to 1.5◦C and the maximum surface temperature reduction could reach 11.5◦C,
Line 43, is HR the same thing as DHR? Please be consistent in the use of acronyms.
Response: Thank you for your advice. The DHR is the same as HR in this paper, thus we made them consistent accordingly.
Lines 65-77 suggest that a study with the exact same objective as this present paper has already been conducted by Reference Sources 12 and 15. Please, be very clear on the contributions that the present paper seeks to make. Otherwise, the paper's novelty becomes very questionable.
Response: Thank you for your valuable comment. The ref. [12] carried out by us is a numerical simulation study for glass bead with different refractive indices (1.5, 1.9 and 2.2), and the ref. [15] done by us is an optical experimental study for developed glass bead RR samples covered with glass sheet, and they are different from the RR materials without glass sheet coating developed in this study. Furthermore, it is considered that the glass sheet surface might affect the RR characteristics of developed RR materials, thus the objective of this paper is different from our previous ones.
Section 2.1, please provide some logistical and/or methodological justification for choosing the colours.
Response: Thank you for your valuable comment. The reason for choosing these colors as reflective layers is that they are commonly used as the reflective layers of RR materials in Japan.
Figure 2 needs to be drawn to scale. To be sure, 3mm can not look exactly as 0.3mm.
Response: Thank you for your valuable comment. We are sorry for our mistake. In fact, Fig.2 is different figure that was used in our previous paper. We have added the right figure is in paper.
Line 130, please provide contextual justification for using the said 30mm distance.
Response: Thank you for your valuable advice. Because the incident light can be well focused on the surface of samples when the distance is set as 30mm. We also added this statement in paper.
Following the methodology for evaluating retro-reflectivity in section 3, the paper needs to provide a methodology for evaluating the cost performance of the proposed materials relative to the commercially available materials.
Response: Thank you for your valuable advice. Because the price of glass beads used in this paper hasn’t been disclosed by collaborative research company, we will do it in our future study.
Line 164, which ones are the 'developed RR samples'?
Response: Thank you for your valuable advice. Samples 1-10 are the developed RR samples.
Line 181&182 - It would appear that this claim was already made by reference sources 12 and 15. Are you simply replicating their work?
Response: Thank you for your valuable comment. In fact, the developed RR materials in this paper are different from our previous ones. The surface of RR materials developed in ref [15] are covered with glass sheet in order to improve the durability of materials. In addition, the process of creating materials has been improved by us in this study. We will show the process in the future.
